# Exploring the Influence of Social Media Information on Interpersonal Trust in New Virtual Work Partners

**Hugo Martinelli Watanuki \*** 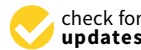 **and Renato de Oliveira Moraes**

Production Engineering Department, Polytechnic School of University of São Paulo, São Paulo 05508-010, Brazil
\* Correspondence: hwatanuki@usp.br; Tel.: +55-119-7245-5049

**Abstract:** This short communication proposes an exploratory investigation regarding the impact of social media information on interpersonal trust in new virtual work partners. The suggested approach assesses this potential impact via a combination of theories from informational economic studies and virtual team research. An initial theoretical model is also proposed.

**Keywords:** interpersonal trust; social media; virtual work

## 1. Introduction

Consider the following scenario in a typical workplace environment: Individual A has been assigned to work with an unknown individual B, with whom interactions will occur exclusively via Information and Communication Technology (ICT) tools. Because the two individuals will not have the opportunity to meet face to face, individual A decides to review individual's B public profiles on social media platforms, such as Facebook, LinkedIn, and Twitter, to know more about the future work partner. Can the information individual A acquires from individual's B public profiles in social media platforms facilitate initial trust development toward individual B? If so, what elements drive this process?

This short communication paper explores these questions. In formal terms, the objective is to propose an exploratory investigation regarding the impact of social media information on interpersonal trust in new virtual work partners. The motivation for this work lies in the possibility of leveraging personal information publicly available on social media platforms to produce positive outcomes in the virtual workplace.

Recent research has shown that the practice of scrutinizing social media profiles to obtain information about individuals has become commonplace among not only friends and family members but also among professionals or even strangers [1,2]. This practice is facilitated by the significant usage of social media platforms by the world's population; as of the second quarter of 2019, Statista [3] shows that more than 2.4 billion people hold an active account in Facebook. Each person's public profile represents a rich source of personal information readily available to any individual around the globe. Whether this information can be useful to facilitate relationship building between new virtual work partners is still mostly unknown [4]; however, this area deserves focused attention as researchers suggest that the majority of active professionals are already working with some form of virtual collaboration [5–7].

From this perspective, one important issue that can be approached is the development of interpersonal trust at the early stages of forming a new virtual relationship [4,8].

Although initial trust is fundamental for effective virtual collaboration because it encourages members to collectively perform transactions and mitigate risk when they interact with each other [4,8,9], the development of interpersonal trust in virtual contexts can be constrained by the lack of physical proximity among individuals [4,8–12]. Therefore, recent research has emphasized the need to

understand what contributes to the initial baseline levels of trust among virtual teammates with no history of collaboration [4,8].

One possible approach to investigate this issue is to focus on the impressions and perceptions individuals form when they are first exposed to publicly available personal information from the new virtual teammate. This is now a common possibility in consideration of the increasing popularity of social media platforms, from which virtual work partners can obtain detailed information about each other, such as personal background, character traits, hobbies, and interests [13–15].

According to Söllner et al. [12], most Information Systems (IS) research on trust has been divided into clusters of studies that focus on trust (i) within virtual teams, (ii) buyer–seller-style relationships in e-commerce, and (iii) among users of online social networks. Several studies have extensively addressed trust-related phenomena between virtual work partners [9–11] or between social media users [15–17]. IS studies attempting to explore the overlap between clusters i and iii in formal organizational settings are more scarce; especially those in which the focus is on the early stage formation of a new virtual relationship. To the best of the author's knowledge, the only study that has explored this specific context is the work of Kuo and Thompson [4]. This study proposed a rudimentary model of initial trust between new virtual work partners based on the social tie information made available by social media platforms. However, their research has not detected significant evidence that this particular information affects trust perceptions between virtual work partners before initial contact has been made. Therefore, ample opportunities exist for a better understanding of the type of social media content that contributes to initial trust development.

This short communication suggests that IS researchers can be more successful in addressing this knowledge gap in trust-related research by using a more comprehensive theoretical framework. To this aim, a combination of theories from virtual team research and information economic studies is leveraged in this study.

## 2. Theoretical Background

This chapter explores the potential inter-effects between the concepts of interpersonal trust and social media technologies in the context of new virtual relationships in the workplace.

### 2.1. Trust in New Virtual Work Partners

Trends like globalization, coupled with advances of ICT tools in recent decades, have pushed companies to move away from a collaboration model based on human resources located within the same physical location. Increasingly, companies have encouraged their employees to collaborate via ICT with virtual partners without their visual proximity [10,13,18], with whom they share no previous work history [4].

When two virtual work partners need to maintain a collaborative relationship, interpersonal trust between them is essential [4,9]. Interpersonal trust is defined by McAllister [19] (p. 25) as "the extent to which a person is confident in, and willing to act on the basis of, the words, actions, and decisions of another". Therefore, in a dyadic relationship, trust involves two specific parties: a trusting party (trustor) and a party to be trusted (trustee).

Interpersonal trust on the trustor side typically develops via a combination of two processes: constructive interactions with the trustee and the assessment of trustee's interpersonal cues that indicate trustworthiness. Whereas the first process tends to contribute to the affective foundations of interpersonal trust, the latter supports its cognitive foundations. Therefore, interpersonal trust is frequently approached as a multidimensional concept [10,11,19,20].

Although important, interpersonal trust between new virtual work partners can be difficult to establish given the constrained context of a virtual relationship. Elements that facilitate trust building during face-to-face interactions, such as social dialogs and opportunities to monitor each other's behavior, may not be present for virtual work partners [4,9,10].

Previous research has suggested that an important prerequisite for the development of interpersonal trust is the trustor's ability to gather information that disconfirms fears that the trustee is not trustworthy [9,15,21]. In this sense, public profiles on social media platforms represent an interesting source of additional information for trustors to assess trustees' characteristics [4].

*2.2. The Effect of Social Media Platforms*

Social media platforms can be conceptualized as an IS artefact consisting of three components: the technological, supporting social interactions; the informational, consisting of user generated digital content; and the social, involving communication and collaboration among people [22,23]. Popular examples of social media technologies are Facebook, Linkedin, and Twitter [8,23,24].

Social media platforms provide individuals with the possibility to exchange information in various forms, comprising not only the user-generated digital content [22,25] but also the perception of social interaction [23,24] which can potentially influence interpersonal trust in real-life relationships [4]. This is justified by the informational cues provided by social media platforms that can be interpreted as signals, as described by the signaling theory from informational economics studies [17]. According to this theory, inequalities in access to information between two parties tend to make the exchange of goods and services between them difficult. Under these conditions, signals that reveal relevant and meaningful information purposefully emanating from one party to the other party can reduce uncertainty and shape a positive behavior toward the other party [17,26,27].

This study suggests that a similar mechanism is applicable to promote trust in a new virtual work partner based on the exploration of his/her public profiles in social media platforms. In this case, positive signals such as identity, presence, reputation, and relationships emanate from the trustee's social media public profiles [28], potentially influencing the trustor's perceptions of trustworthiness.

Such a diversified set of signals must require an equally diversified set of theories to account for their effects on interpersonal trust. Although previous research [18] has suggested that virtual collaboration can be approached from different theoretical perspectives in the IS domain, the same study has shown that three theories have been most frequently leveraged to explain social aspects of virtual teams: social presence theory, social information processing theory, and social identity or deindividuation theory. Given the focus of this study on the social aspects of a new virtual relationship, these are the three theories that were selected for further analysis.

First, social presence theory (SPT) [18,29] suggests that the awareness of other social participants' interactions (i.e., social presence) can be augmented in communication via ICT tools as more channels become available for the expression of nonverbal cues. A high degree of social presence is important for the development of trust because the trustor's perception of human interactions with the trustee is a precondition for trust [30], especially its affective dimension [11,19,20]. Despite the limited presence of actual human contact in virtual workplace environments, research has suggested that signals of social presence can be embedded in technology artefacts, such as websites, as well as via images and biographical information that convey sense of personal and sensitive human contact [11,20,24,30]. This is in agreement with the informational component of social media technologies, the focus of which is on user-created content, such as personal profiles, text, photographs, and video streams [22,23].

Second, social information processing theory (SIPT) [18,31] proposes that, when communicating solely via ICT tools in which nonverbal cues are not available, individuals adapt and use available information to form impressions and evaluate others. Therefore, SIPT suggests that, in virtual environments, people tend to rely on peripheral social information, such as language, written attitude, and self-disclosure to form impressions about others [24,25,31]. In this sense, social media technologies provide its users with generous identity signals to disclose information about other individuals [22,23]. How these signals affect different dimensions of interpersonal trust depends on whether they make salient aspects of personal identity or social identity [32]. For instance, with regard to personal identity, research has suggested that personal identity signals, such as the availability of an individual's work history information on a social media profile, can function as a set of cues that allow others to

better evaluate this individual's professional credentials [24,25], which, in turn, can help to foster a cognition-based component of trust towards him/her [10,11,19].

With regard to social identity, according to the social identity or deindividuation (SIDE) theory [18,33], in contexts where individuating cues about others are limited, individuals categorize themselves as part of social groups based on the information made available by other sources. Therefore, when a trustee's signals of shared social identity with the trustor are available in a social media profile, such as common interests, experiences, values, and demographic traits, these signals may accentuate the perception of similarity between them, enhancing the trustor's feelings of attraction and identification toward the trustee [30,32]. These are elements that can help foster both affective and cognition-based components of trust [10,16,19].

The proposed relationships described above are illustrated in Figure 1.

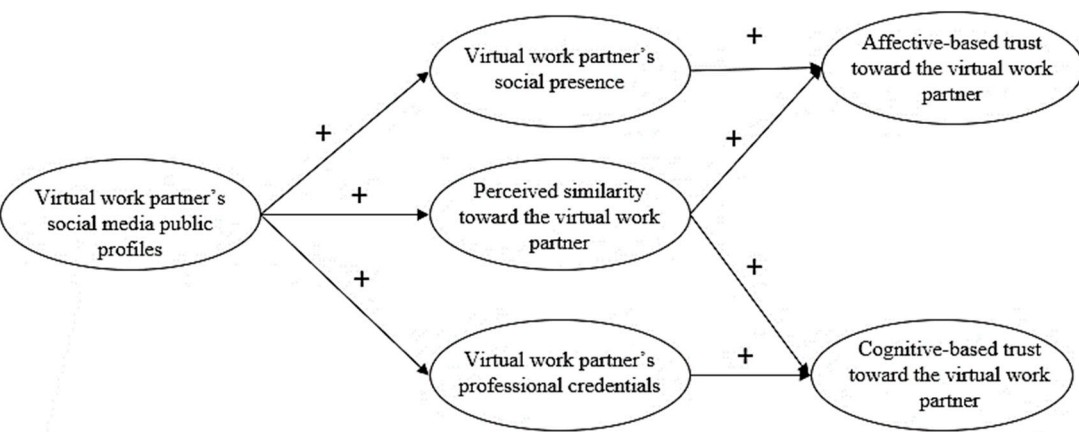

**Figure 1.** The theoretical model proposed.

## 3. Concluding Remarks

This short communication paper proposes an exploratory investigation regarding the impact of social media information on interpersonal trust in virtual work partners. By considering a wider theoretical framework in comparison with previous studies, an initial set of relationships have been proposed.

The theoretical model presented suggests that social media information can provide important signals that contribute to the initial development of interpersonal trust in new virtual work partners. As a result of their defining characteristics and constituent elements, social media technologies can help increase an individual's perception of a virtual work partner's social presence, perceived similarity, and professional credentials, leading to increased affective and cognitive-based trust toward the new virtual work partner.

From a theoretical standpoint, it is expected that the alternative approach proposed by this short communication will help to increase the chances of IS researchers to address previous inconclusive findings regarding the impact of social media information on the development of interpersonal trust in new virtual work partners [4]. Furthermore, this study can promote a better understanding of the type of social media content that contributes to initial trust development. From a practical perspective, this study can provide practitioners with an increased perception about the importance of disclosing quality information in their public social media profiles as well as managing online reputation for improved future virtual work relationships.

Given the exploratory nature of the theoretical model presented, its further development is encouraged via the inclusion of potential moderating and control variables.

One potentially important moderating variable is the concept of propensity to trust or the general willingness that an individual possesses to trust others [34]. According to Kuo and Thompson [4], in the absence of information about the trustee, trustors have little or no basis on which to assess the

trustee's trustworthiness. In such situations, trustors with increased propensity to trust are expected to engage in trusting behaviors because they are especially inclined to trust other individuals.

Another potential control variable to be considered in this model is an individual's gender. According to recent research from Sun et al. [35], due to the inherited differences in social behavior between females and males, the trust-building mechanism in social media contexts varies across gender. Specifically, males may give more emphasis on competence-based factors to build trust whereas females may rely more on emotional or affective factors.

Discussions regarding the empirical validation of the theoretical model presented here are also necessary and constitute an important opportunity for future research. One viable alternative may be conducting surveys among business professionals. In this case, and in line with previous research [4], a hypothetical scenario can be presented to survey participants in which they are asked to evaluate the perceived trustworthiness of a potential new virtual work partner. The survey participants can then be exposed to fictious social media public profiles with different levels of personal information quality and volume (i.e., different signal levels), and have their perceived trustworthiness levels assessed.

Finally, a word of caution is required regarding the practice of exploring personal information from public social media profiles; social network users tend to be concerned about their privacy [13,36]. Social media users are generally willingly to share their identities; however, they are also concerned about the usage of their information by unknown others [7,28]. To circumvent privacy concerns, social media users can sometimes develop identity strategies, such as creating virtual identities that differ from their real identities or abandon their social media accounts [36]. These are challenges that need to be considered in future development of this research.

**Author Contributions:** Conceptualization, H.M.W. and R.d.O.M.; Writing—Original Draft Preparation, H.M.W.; Writing—Review & Editing, H.M.W.; Visualization, H.M.W.; Supervision, R.d.O.M.

**Funding:** This research received no external funding.

**Acknowledgments:** The authors would like to acknowledge Carlos Alberto Vanzolini Foundation essential financial support, which enabled this work.

**Conflicts of Interest:** The authors declare no conflict of interest.

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
