# Peer review of "Exploring the Influence of Social Media Information on Interpersonal Trust in New Virtual Work Partners"

_informatics, doi:10.3390/informatics6030033_

Round 1
Reviewer 1 Report
The premise of this paper is that most work on computer-mediated trust focuses (1) on virutal teams, (2) on buyer-seller relationships in e-commerce, or (3) social network users. They claim that too infrequently have works attempted to bridge (1) and (3), looking at how virtual teams interactions on social networks impact trust specifically.
I am not an expert on research on virtual teams, so I cannot judge whether or not this work cites all relevant studies. (Though I find it hard to believe that only two studies discuss social media use among virtual work partners; are there any qualitative studies on social media use among distributed teams that may indicate effects on trust?).
My main issue with this paper is that the authors neglect to spell out exactly why this intersection is appealing for further research. They explain that trust is important, especially in virtual work partners. They also establish that social media is important with regard to trust (or at least get most of the way there). However, they don't quite tie it all together. Why social networks and virtual work partners?
In general, it is very difficult to assess the significance or interest of this paper, as the reader is unsure about the authors' motivations for performing this work. This effect is confounded by the concluding remarks, which don't tell us exactly what the authors plan to do, or what kind of impact they expect their work will have.
Author Response
Dear Reviewer,
Thank you very much for the attention dedicated towards our manuscript and for the recommendations. There has been a significant effort from the authors to incorporate the items raised and we hope to have satisfactorily attended all of them in an effort to improve the quality of the paper. The revised manuscript has been submitted and the responses to the specific recommendations made are provided attached.
Once again, thank you for considering our revised manuscript.
Sincerely,
The authors

Reviewer 2 Report
Thank you for the opportunity to comment on your manuscript. In general, I appreciate that you stress the importance of sound theory to examine and predict effects of (social) media on interpersonal trust. However, I would have wished to see an overarching framework justifying the selection of the three theories (and not others). Moreover, I found the constructs not precisely defined. In fact, facebook, Linkedin and Twitter are quite different platforms (e.g., some are work-related, some are not, media richness differs a lot, etc.) so that I was wondering to which media aspects you refer to more precisely. On the other hand, you seem to equate ICT with lack of information - but that is not always true. What about more recent developments in virtual collaboration, such as Web-conferencing or Virtual Reality tools? Another shortcoming of the paper is that more recent research on technology and trust is not covered. And finally, I would expect also some thoughts on aversive effects of social media usage in this respect: Personally, I would not be amused if I find out that my colleagues are checking my private(!) facebook account…
Together, although I like that you stress the importance of sound theoretical work, just listing three theories that might be relevant for media effects on interpersonal trust is not a very strong contribution to the literature, I'm afraid. Clarifying the open questions noted above would probably make a stronger case, together with more precise explanations of the assumed links between the constructs (e.g., why should ”The way these additional cues can affect interpersonal trust … depend if they make salient aspects of personal identity or social identity”? p. 2) and how the described assumptions can be tested empirically.
Author Response

(The authors gave the same response as above.)

Reviewer 3 Report
This short communication deals with an interesting subject, the effects of social media on trust between virtual work partners. The focus of the paper is clearly presented in the introduction, the theoretical background section presents a good overview of the bibliography on trust in virtual work partners, and describes three social theories for the potential influence of social media that have been used in a proposed theoretical model. The paper is well presented and develops a clear line of argument towards the proposed future research.
The paper can be improved if:
a) you elaborate more on the impact of their proposed research. In the paper you state that it "will help increase the chances of IS researchers to address previous inconclusive or limited findings..". Please explain a bit more why are these findings inconclusive (what is missing) and how is your proposed work expected to fill the gap?
b) you give more details about the design and methodology of the exloratory investigation you propose. E.g. name some 'potential moderating and control variables'.
Finally, it is not clear to me if "the information individual A acquire[s] via social media technologies" stated in the introductory scenario refers to public social media profiles (e.g. the page of a user in LinkedIn), or private updates / posts available to friends and followers.
A minor correction: on p.3. line 3 'identify' -> 'identity'
Author Response

(The authors gave the same response as above.)

Round 2
Reviewer 1 Report
This paper is much improved from its prior form; it engages significantly with relevant literature and carves out a space which appears to be novel (facilitating trust among distributed work partners through what we may colloquially call "stalking" their social media profiles).
My one suggestion for the paper in its current form would be to draw on data, if available, that this practice is commonplace enough to warrant specific investigation. What this paper appears to be missing currently is urgency around this particular topic of discussion. Perhaps distributed work is becoming more common---certainly there are statistics around this. Perhaps surveys or even market research show the practice of using social media profiles to make judgments about others is becoming more widespread. Some "hook" to demonstrate the relevance of this research project would be helpful.
I commend the authors for their effort in improving this work.
Author Response
23-Aug-2019
Dear Reviewer,
Thank you very much for the attention dedicated towards our manuscript and for the recommendations. There has been a significant effort from the authors to incorporate the items raised and we hope to have satisfactorily attended all of them in an effort to improve the quality of the paper. The revised manuscript has been submitted and the response to the specific recommendations made by the reviewer are provided below:
Reviewer´s comment: This paper is much improved from its prior form; it engages significantly with relevant literature and carves out a space which appears to be novel (facilitating trust among distributed work partners through what we may colloquially call "stalking" their social media profiles).
My one suggestion for the paper in its current form would be to draw on data, if available, that this practice is commonplace enough to warrant specific investigation. What this paper appears to be missing currently is urgency around this particular topic of discussion. Perhaps distributed work is becoming more common---certainly there are statistics around this. Perhaps surveys or even market research show the practice of using social media profiles to make judgments about others is becoming more widespread. Some "hook" to demonstrate the relevance of this research project would be helpful.
I commend the authors for their effort in improving this work.
Author´s response: The authors have conducted an additional literature review to identify data to support the relevance of the research project. As a result, the data from six additional references were added to the manuscript highlighting not only the impressive numbers regarding social media usage by the world´s population; but also, the increasing trends toward the usage of virtual work by the organizations, and the practice of scrutinizing social media profiles by the general population. This supporting data has been compiled and presented as an additional paragraph in the introduction chapter of the manuscript (3rd paragraph).
Once again, thank you for considering our revised manuscript.
Sincerely,
The authors